# Effect of Antioxidant and Antimicrobial Coating based on Whey Protein Nanofibrils with TiO_2_ Nanotubes on the Quality and Shelf Life of Chilled Meat

**DOI:** 10.3390/ijms20051184

**Published:** 2019-03-08

**Authors:** Zhibiao Feng, Lele Li, Qiannan Wang, Guangxin Wu, Chunhong Liu, Bin Jiang, Jing Xu

**Affiliations:** College of Science, Northeast Agricultural University, Harbin 150030, China; fengzhibiao@neau.edu.cn (Z.F.); LILELE931022@163.com (L.L.); W18346111097@163.com (Q.W.); angeloxin@yahoo.com (G.W.); jiangbin@neau.edu.cn (B.J.); xujing@neau.edu.cn (J.X.)

**Keywords:** protein nanofibrils, edible coating, antioxidant activity, antimicrobial activity

## Abstract

Whey protein nanofibrils (WPNFs) can be used in edible films and coatings (EFCs) because of its favorable functional properties, which rely on its well-ordered *β*-sheet structures, high hydrophobicity, homogeneous structure, and antioxidant activity. In the present study, WPNF-based edible coatings with glycerol (Gly) as plastic and titanium dioxide nanotubes (TNTs) as antimicrobial agents were studied. TNTs not only showed greater antibacterial activity than titanium dioxide nanoparticles (TNPs), but also increased interactions with WPNFs. The WPNF/TNT film had a smooth and continuous surface and was homogeneous with good mechanical properties. WPNF/TNT edible coatings (ECs) can help improve lipid peroxidation and antioxidant activity, limit microbial growth, reduce weight loss, and extend the shelf life of chilled beef. Given that the WPNF/TNT film components are low cost and show high antioxidant and antimicrobial activity, these optimized films have potential applications for various food products, including raw and chilled meat.

## 1. Introduction

Edible films and coatings (EFCs) degrade naturally, because they are normally based on natural ingredients, such as proteins [1,2], polysaccharides [3], and lipids [4]. They often have many functions, such as preventing weight loss, reducing lipid oxidation, providing microbial stability to foods, protecting antimicrobial compounds against adverse reactions (such as oxidation or hydrolysis), and preserving functional foods during processing and storage [5,6]. As lipid oxidation and bacterial growth are the two main factors that determine food quality loss and shelf life reduction [7], EFCs used in food products that inhibit lipid oxidation and microbial growth to extend their shelf life have attracted increasing attention from scholars. 

Protein-based EFCs are attractive for application to foods because of their excellent sensory properties and potential to protect food products from the surrounding environment [8,9,10,11]. However, the hydrophilic nature of whey protein edible coatings (ECs) causes them to be less effective as moisture barriers [12]. After whey protein isolates (WPIs) self-assemble into semiflexible amyloid-like protein nanofibrils (WPNFs), properties, like gelation, emulsification, hydrophobicity, foam stability, rheologic properties, and especially antioxidant activity, significantly improve [13]. Further, WPNFs are suitable for use in food packaging to protect foods [14]. 

Titanium dioxide (TiO_2_) is a non-toxic, inexpensive, and photo catalytic sterilization [15] approved by the United States Food and Drug Administration (FDA) for use in human foods and food contact materials [16]. TiO_2_ can meet hygienic design requirements in food processing and packaging surfaces because of its self-disinfecting properties. TiO_2_ nanotubes (TNTs) show greater antibacterial activity than TiO_2_ nanoparticles (TNPs) against a wide spectrum of cells and organisms, such as bacteria, fungi, and cancer cells, owing to their special electric and mechanical properties, high photo catalytic activity, large specific surface area, and high pore volume [16]. When TiO_2_ is incorporated into a biopolymer matrix, it can help decrease the transmittance of light’s visible, ultraviolet (UVA, wavelengths of 320 to 400 nanometers) and mid-ultraviolet (UVB, wavelengths of 280 to 320 nanometers) spectra [17]. Therefore, TNTs might be a good choice for use in food packaging systems to prevent spoilage from light-induced oxidation.

Therefore, the objective of this study was to develop WPI into WPNFs for TNT- and Gly-incorporated packaging to enhance the quality of chilled meat (beef) and to evaluate the effect of the films with respect to antioxidant and antimicrobial activities. Micrographs of the coating solution, microstructure, ultraviolet spectra, mechanical properties, moisture content (MC), water solubility, transparency, water vapor permeability (WVP), and antimicrobial activity of films, as well as the main parameters of antioxidant and antimicrobial activities in chilled beef stored at 4 °C, were monitored.

## 2. Results

### 2.1. Transmission Electron Microscopy (TEM) Micrographs of the Coating Solution

TEM of negatively stained samples is an essential method for determining the morphology of fibrils. Figure 1 shows the TEM images of WPNFs at pH 2.0 and 7.0, TNTs, and WPNF-TNTs at pH 7.0. Linear, long, and unbranching fibrils were observed in the micrographs of WPNFs at pH 2.0 (Figure 1A), which we described in our previous report [14]. With a pH adjustment to 7.0, the fibrils became shorter (Figure 1B). Fibril emulsions are expected to be more stable because of their higher viscosity and faster migration of fibrils to the interface [18]. TNTs have a specific surface area (Figure 1C), which leads to TNTs with different physical and chemical properties, such as photo catalytic and mechanical behavior [19]. TNTs and WPNFs interact well in solution (Figure 1D). 

### 2.2. Texture Profile Analysis

#### 2.2.1. Scanning Electron Microscopy (SEM) of the Microstructure of Films

The microstructure of the surface and cross-sections of the films from WPI or WPNFs with TNTs was examined by scanning electron microscopy as shown in Figure 2. Both films had a smooth and homogenous surface without cracks or pores, but they appeared different in the cross-sections. The cross-sections of the WPI/TNT films were rough (Figure 2A), with some irregular particles uniformly distributed, and the WPNF/TNT films had a smooth and continuous surface, presenting a homogeneous, compact, and cohesive interior without grains or pores (Figure 2B). This second profile is characteristic for most one-component hydrocolloid films and indicated that the film was formed with an ordered-phase and homogeneous network structure [14]. 

#### 2.2.2. Light Transmission and Film Transparency

A conventional spectrophotometer equipped with an integrating sphere is often used to measure reflectance or transmittance to assess for a smooth surface, which allows the physical status of a film to be monitored [20,21]. Transmittance measurements on small samples are much more accurate than reflectance measurements [20]. Therefore, the ultraviolet spectroscopy (UV)-Vis (200–800 nm) transmission spectra of different composite films were obtained (Figure 3A).

Coating transparency is very important for consumer acceptance and product appearance [22]. The films showed very high transmittance (above 80%) in the visible range (380–650 nm), with the WPNF/TNT film showing the highest transmittance. Compared with the visible region, the transmission of the films with TNPs and TNTs was very low in the UV region of the light spectrum. Compared with the films containing TNPs, the absorbance of the films with TNTs was higher in the UV region. 

#### 2.2.3. Physical Properties of Nanocomposite Films

##### Mechanical Properties

Mechanical properties are important indices for evaluating the physical properties of edible films. Table 1 shows the evolution of the mechanical properties of the films. There was no significant difference in the thickness between the EFs, indicating that the formation of WPNFs and the addition of TNPs or TNTs did not influence their thickness (*p* < 0.05), but the tensile strength (TS) and elongation at break (EB) of the films were influenced by these factors. With the addition of TNPs or TNTs, TS was significantly increased. The EB values of the films decreased (*p* < 0.05) with the addition of TNPs, but that of the films had not obviously changed (*p* < 0.05) with the addition of TNTs. This may have been caused by large TiO_2_ agglomerates that made the bond matrix discontinuous and led to catastrophic failure of the film elongation [17,23,24]. These indicated that TNTs have many potential applications as food packaging materials.

##### Moisture Content (MC), Solubility in Water, Transparency, and Water Vapor Permeability (WVP)

The MC, solubility, transparency, and WVP of EFCs are four important parameters of EFCs as shown in Table 2. The mean MC value is related to the total void volume occupied by water molecules in the network microstructure of a film. Solubility is a significant functional property of EFCs based on the hydrophilicity of a material. The transparency of EFCs has a direct effect on the coated product appearance and is key to acceptance by consumers [22]. WVP is used to measure permeability and the difficulty of water passing through a material. 

Because the WPNFs had well-ordered *β*-sheet structures, higher hydrophobicity, and a homogeneous network structure [14], WPNF-based films showed significantly lower mean MC, lower solubility, higher transparency, and lower WVP values than the WPI-based films (*p* < 0.05) (Table 2). The addition of TNPs or TNTs changed the functional properties of EFCs. First, the MC values of the films were higher than the films without them, which may result from the super-hydrophilicity of TNPs [20]. Second, the solubility in water values of the films were lower than the films without them. This suggested that the addition of TNPs resulted in greater hydrophobicity, decreasing access for water molecules to the protein. WPNF/TNT films showed the lowest solubility value (26.3 ± 1.5, *p* < 0.05), indicating that TNTs are more able to combine with WPNFs than TNPs, leading WPNF/TNT films to absorb less water and have greater hydrophobicity. Third, the addition of TNPs or TNTs decreased the transmittance of the films, as reported previously [23]. What is interesting is that the transparency of the TNTs films was higher than that of films with TNPs. Lastly, the WVP of films showed that permeability with the addition of TNPs or TNTs was insignificantly reduced (*p* > 0.05). The possible reason for this phenomenon could be the increased hydrophobicity of TNPs or TNTs [25]. These results indicate that TNTs are more able to combine with WPNFs than TNPs, causing the WPNF/TNT film to have a lower mean MC, lower solubility, and higher transparency. Therefore, these films may be more suitable for food packaging.

#### 2.2.4. Antimicrobial Activity

The antimicrobial activity of WPNF-based and WPI-based film was evaluated using the disc diffusion method (Figure 3B). We selected two gram-positive bacteria (*L. monocytogenes* and *S. aureus*) and two gram-negative bacteria (*S. enteritidis* and *E. coli*) common in meat products. The inhibition zones were always more than 10 mm, which was the diameter of the film strip. If there was no clear area surrounding the strip, the value would be defined as zero, which was observed with the WPI films. Films without any antimicrobial compound have no inhibitory effect on the tested microorganisms; interestingly, the WPNF films had a very small inhibition zone, which might have been due to the antioxidant ability of WPNFs.

Each film had the largest zone of inhibition against *L. monocytogenes*, with no significant differences between the inhibitions of the other three bacteria. The inhibition zones of the WPI/TNP and WPI/TNT films were larger than those of the WPI film. WPNF-based groups showed similar results, indicating that the antimicrobial effect was mainly due to TNPs and TNTs. The inhibition zones of the TNT-based films were larger than those of the TNP-based films.

### 2.3. Functional Properties of the Ecs on the Preservation of Chilled Meat

#### 2.3.1. Lipid Oxidation

The thiobarbituric acid reactive substances (TBARS) concentrations in meat, which represents its malonaldehyde concentration, are presented in Figure 4A. The concentrations of TBARS in control samples were significantly increased at 0–15 days and reached 2.52 mg MDA/kg at the end of the storage period. There were no significant differences (*p* > 0.05) between the WPI-based samples, and the trends over time were similar to those in the control samples. This indicated that the incorporation of TNPs or TNTs into the coatings did not increase the protection of meat samples against lipid oxidation. 

The concentrations of TBARS in all WPNF-based samples showed no significant difference (*p* > 0.05). They were stable in these samples during the first 9 days of storage, increased significantly in the last days, and then finally reached 0.94 mg MDA/kg. Moreover, the TBARS values of WPNF-based samples (0.94–1.03 mg MDA/kg) were significantly lower than those of the control and WPI-based samples (2.33–2.52 mg MDA/kg) during storage, and lower than the threshold for the rancidity of beef (2.28 mg MDA/kg) [26]. WPNF-based samples significantly reduced the increase in the TBARS value during storage, slowed the occurrence of lipid oxidation, and kept the meat fresh. These showed that WPNFs, instead of TNPs or TNTs, played a major role in preventing lipid peroxidation and therefore are more suitable for use in coatings than WPI for food preservation.

#### 2.3.2. DPPH Radical-Scavenging Activity

The DPPH radical-scavenging activity of WPNF-based samples was greater than that of WPI-based samples on the same day (Figure 4B). No significant differences in WPI/TNP and WPI/TNT samples were found compared to those with WPI, and WPNF-based samples showed the same results. The results indicated that the antioxidant activity may be attributed to WPNFs, as the addition of TNPs and TNTs clearly did not affect the antioxidant activity of the samples, as reported previously [14]. Combining the TBARS values and DPPH radical-scavenging activity suggests that WPNFs could be utilized to prevent the oxidation of food products.

#### 2.3.3. Antimicrobial Activity

Microbial growth leads to discoloration, textural changes, and off-flavor development, which subsequently reduces shelf life [27]. The bacterial TVC is one of the most important indexes used to evaluate the quality and safety of meat, and it is the quantitative sanitary standard used to identify the conditions and degree of meat contamination [28]. The TVC values of all samples increased with storage time at 4 °C (Figure 4C). The initial TVC of the fresh meat was ca. 4.0 lg CFU/g, indicating good quality meat. A value of 7.0 lg CFU/g or more is considered the upper microbiological limit for good quality fresh beef.

The TVC values of the control samples rose quickly from 4.0 lg CFU/g to 8.6 lg CFU/g over 15 days of storage. When stored for 9 days, the TVC reached 6.9 lg CFU/g, and the beef exceeded the acceptable microbiological content level. For WPNF/TNP-coated samples, the TVC values were slower, reaching 6.9 lg CFU/g after 15 days of storage. Benefiting from super photo catalytic bactericidal activity, the TVC values of WPNF/TNT-coated samples showed the slowest increase, rising to 5.7 lg CFU/g after 15 days and extending the shelf life of chilled beef compared with that of the control group.

#### 2.3.4. Weight Loss

Water loss, which affects the appearance and quality of meat, takes place mainly during storage. There was a significant difference in this loss between the two groups (Figure 4D). The WPNF/TNT coating efficiently preserved the weight of the meat, allowing a mean reduction in weight loss of 7.87% over 15 days of storage, compared with a loss of approximately 12.67% in the control sample. In addition to forming a physical barrier, the lower water vapor permeability might also have contributed to the lower weight losses during storage. 

## 3. Discussion

The morphology of fibrils by TEM of negatively stained samples (Figure 1) showed that TNTs and WPNFs can interact well in solution. When the solutions were adjusted to 7.0, the surface charge density, content of exposed hydrophobic groups, and rheological and interfacial properties of the protein change [18,29], and the surface energy of TNTs is very high, providing more opportunities for interaction between TNTs and WPNFs. The surfaces and cross-sections of the WPNF/TNT films (Figure 2) with a more smooth and continuous surface have also confirmed this result.

Due to the well-ordered *β*-sheet structures, higher hydrophobicity, and homogeneous network structure of the WPNFs films [14], the WPNF-based films exhibited better mechanical properties (Table 1), lower mean MC, lower solubility, higher transparency, and lower WVP values (Table 2). TS were significantly increased with the addition of TNPs or TNTs. This may be because of the interactions between the carboxylic and sulfhydryl groups in certain amino acids with TiO_2_ [24]. TNPs and TNTs can be embedded into a network of proteins in suitable conditions, resulting in excellent mechanical properties of polymers through electrostatic attraction and hydrogen, O-Ti-O, and protein-TiO_2_ bonding [17]. These results coincide with the highly homogeneous structure observed in the SEM micrographs. The EB values of films decreased (*p* < 0.05) with the addition of TNPs. This may have been caused by large TiO_2_ agglomerates that made the bond matrix discontinuous and led to catastrophic failure of the film elongation [17,23,24]. However, the EB values of films did not obviously change (*p* < 0.05) with the addition of TNTs, because TNTs may be more able to combine with WPNFs than TNPs. WPNF/TNT films absorbed less water, and had greater hydrophobicity and higher transparency than that of WPNF/TNP films, indicating that TNTs have many potential applications as food packaging materials.

TiO_2_ is a photocatalyst and is widely utilized as a self-cleaning and self-disinfecting material for surface coatings in the food and environmental industry. Hydroxyl radicals (•OH) and reactive oxygen species (ROS) generated on the illuminated TiO_2_ surface play a role in inactivating microorganisms by oxidizing the polyunsaturated phospholipid component of the cell membrane of microbes [16]. Transmission of the films with TNPs and TNTs was much lower in the UV region than the visible region, because UV absorption around 300 nm was partly attributed to the aromatic amino acid residues, tyrosine (Tyr) and tryptophan (Trp), in the protein and partly attributed to the TNPs and TNTs. The absorbance of the WPNF/TNT film in the UV region was higher than the WPNF/TNP film. Because TNTs have larger specific surface areas and a higher refractive index than TNPs [30,31], they can transport electrons from the excited Tyr and Trp residues to the particles themselves, which can enhance the UV absorption of these groups [17]. This ability of TNPs and TNTs made the films absorb UV light and avoid food deterioration, which are more intensive with and easily caused by UV light [32]. The inhibition zones test (Figure 3B) confirmed that the addition of TNPs and TNTs increased the antimicrobial activity of the films. TNT-based films showed stronger antimicrobial activity than those of the TNP-based films because TNTs have a higher interfacial charge transfer rate and excellent photo catalysis.

During meat storage, weight loss, lipid oxidation, and bacterial growth influence the product’s quality and shelf life. These processes all limit the shelf life to several days under normal conditions. The semipermeable barrier provided by TNT-incorporated ECs extends the shelf life by reducing moisture, solute migration, oxidative reaction rates, and the bacterial growth rate. Functional properties of the ECs on the preservation of chilled meat showed that WPNFs played a major role in enhancing the protection of meat samples against lipid oxidation and DPPH radical-scavenging activity as reported previously [14]. The addition of TNPs or TNTs enhanced the antimicrobial activity, and TNTs showed greater antibacterial activity than TNPs, owing to their special electric properties, high photo catalytic activity, large specific surface area, and high pore volume [16]. Therefore, the WPNF/TNT film with high antioxidant and antimicrobial activity showed great potential for application to various food products, including raw and chilled meat.

## 4. Materials and Methods

### 4.1. Materials

WPI (>91.5%) was purchased from Hilmar Industries (Hilmar, CA, USA). TNTs (purity > 99.9%, OD ≈ 20 nm) and TNPs were obtained from the Beijing Dk Nano S&T, Ltd. (Beijing, China). 2-Thiobarbituric acid (TBA), trichloroacetic acid (TCA), and malonaldehyde bis (dimethyl acetal) were purchased from Sigma-Aldrich (St Louis, MO, USA). 2,2-Diphenyl-1-picrylhydrazyl (DPPH) was purchased from Aladdin Reagent Co. (Shanghai, China). Pathogenic bacteria were obtained from BNCC Biological Technology Co., Ltd. (Nanjing, China). Ultrapure water was prepared from a Milli-Q water purification system (Millipore, Billerica, MA, USA) and was used throughout the experiments. Other reagents obtained in this study were of analytical grade.

### 4.2. Preparation of WPNFs

WPNFs were prepared according to the method described in our previous report [14]. WPI was dissolved in deionized water at 5% (w/v). The pH of the solution was adjusted to 2.0 with 3 M of hydrochloric acid, and the mixture was stirred at 25–28 °C for 30 min, then centrifuged at 9000× *g* for 15 min at 4 °C (Z236HK Hermle, Wehingen, Germany). The supernatant was vacuum-filtered through a fiber membrane (0.45 µm pore size, Aladdin, Shanghai, China) to remove undissolved protein. The filtrate (WPI solution) was used as a control in the following experiment. WPNFs were obtained by incubating the filtrate at 80 °C for 10 h with 220 rpm constant magnetic stirring.

### 4.3. Preparation of the Edible Coating Emulsions

To obtain coating emulsions, TNTs (1% w/w) or TNPs (1% w/w) as the antibacterial agents, and glycerol (4% w/v) as a plasticizer were added to WPNF (5% w/v) solution with constant gentle stirring at 90 °C for 30 min in a water bath. After this, the solutions were adjusted to pH 7.0 with the addition of 2 M NaOH and stirred continuously using an ultrasonic homogenizer (SB-800 DTD Xinzhi-biotechnologies Inc., Ningbo, China), vacuumed to eliminate dissolved air, and then cooled to room temperature in an ice bath. In addition, a WPI (5% w/v) solution was prepared as a control using the same procedure.

### 4.4. Characterization and Physical Properties of Nanocomposite Films

#### 4.4.1. Negative Stain Transmission Electron Microscopy (TEM)

WPNF samples were ultrafiltered using a method reported previously to reduce the background in TEM images [33]. The coating solution was diluted approximately 20 fold, transferred to a special copper mesh on a carbon film, allowed to stand for 20 min, and then touched against filter paper to wick away excess sample. A droplet of 2% uranyl acetate was then added to the dried copper, the copper was held for 8 min, and any excess solution was removed as before. Electron micrographs were obtained by TEM (H-7650 transmission electron microscope, Hitachi, Tokyo, Japan).

#### 4.4.2. Scanning Electron Microscopy (SEM)

The structures of the films were measured using SEM (Hitachi S-3400N, Tokyo, Japan) according to the method of Fabra et al. [34] with some modifications. Film samples were cryo-fractured by immersion in liquid nitrogen and then mounted perpendicularly to the surface of bronze stubs. The fracture surfaces (cross-section) of the films were then sputtered with a thin layer of gold. A silicon probe (Hitachi, Tokyo, Japan) with a cantilever length of 125 mm and a resonant frequency of approximately 500 kHz was used. The scan rates were between 0.5 and 1.0 Hz [35].

#### 4.4.3. Ultraviolet Spectroscopy

The UV/vis spectra of the nanocomposite were recorded over a wavelength range from 200 to 800 nm using a UV spectrophotometer with an integrating sphere attachment (Shimadzu, Kyoto, Japan) [36]. Each film specimen was cut into a rectangle and placed directly in a spectrophotometer test cell. Baseline correction was performed using a barium sulfate whiteboard. At least three replicates were measured.

#### 4.4.4. Physical Properties of Nanocomposite Films

To examine the film-forming state, the film-forming emulsions were made into films. Polypropylene plates (15 cm diameter) were employed as supports to prepare the films and conditioned in an environmental chamber (Blue-Pard Pharma, China) set at 40 °C for 24 h. The dried films were peeled off and stored in a desiccator at relative humidity (RH) 48 ± 5% and 25 °C for 24 h.

##### Thickness Measurement

The samples were conditioned at 25 °C for 24 h, and the thickness of the edible films (EFs) was measured (exactness of ± 0.001 mm) using a digital external micrometer (Mitutoyo Co., Kawasaki, Japan) at 10 different points of the film. Averaged thickness values were obtained and used in all calculations.

##### Mechanical Properties

Tensile strength (TS) and elongation at break (EB) were determined using a TA Plus Texture Analyzer (SMS TA, Surrey, UK) according to the method of He et al. [37] with a slight modification. The films were cut into 2 cm × 5 cm strips. The films were held parallel with an initial grip separation of 3 cm and then pulled apart at a head speed of 25 mm/min. The maximum force (*F*_max_, N) at film breaking and the film length (*L*, cm) from tension to breaking were recorded. TS (MPa) and EB (%) were calculated using the following formulas:$$\mathrm{TS}(\mathrm{MPa})=\frac{F_{\max}}{S}$$
$$\mathrm{EB}(\%)=\frac{\Delta L}{L_{0}}\times 100$$where *S* is the cross-sectional area (m^2^) of a sample, Δ*L* is the length of a film at the moment of rupture, and *L*_0_ is the original grip length of the films.

##### MC Measurement

The film samples were cut into pieces (200–300 mg) and immediately weighed and dried for 24 h in an air-circulating oven at 105 °C. Then, the film samples were conditioned at 50% RH and 25 °C for 24 h and weighed to determine the MC. MC values were determined five times as the percentage of the initial film weight lost during drying.

##### Measurement of Solubility in Water

Solubility of the films was determined according to a previously described procedure [38] with modifications. Briefly, pieces of films (50 mm × 50 mm) were dried in an oven at 80 °C until a constant weight to obtain the initial film dry weight. The piece of film was then placed into a test tube with 100 mL of distilled water. Then, the tubes were placed on a shaking platform for 24 h at 25 °C. After immersion, the swollen samples were removed and dried again in the oven at 80 °C until a constant weight to determine the final dry weight.

##### Transparency Measurement

The transparency of the films was determined according to a previously described procedure [39]. The percentage of light transmission (T%) through the films was measured using a spectrometer (UV-2500, Shimadzu, Tokyo, Japan). The film samples were cut into strips (4 cm × 1 cm) and attached to one side of a spectrophotometer cell; an empty cell was used as a control. The relative transparency of films was measured at 600 nm and calculated using the following formula:$$\mathrm{Transparency}\%=\frac{A_{600}}{x}\times 100$$where A_600_ is the absorbance at 600 nm, and *x* is the film thickness (mm). At least five strips of each film type were tested.

##### Water Vapor Permeability (WVP)

WVP was determined according to the ASTM method with some modifications [40]. A film with a diameter of 40 mm was placed on top of a flat-lipped glass beaker containing distilled water and wax sealed at the bottleneck to maintain 95% RH. Then, the beaker was placed in a desiccator with silica gel at 25 °C and 1.5% RH for 48 h to obtain the constant vapor pressure difference between the films’ internal and external sides. The WVP was then calculated as follows:$$\mathrm{WVP}=\frac{\Delta w\times l}{A\times \Delta t\times \Delta p}$$where Δw/Δ*t* is the weight of moisture loss per unit time (g·s^−1^), *l* is the film thickness (m), *A* is the area of the exposed film (m^2^), and Δ *p* is the water vapor pressure difference between the sides of the film. Three replicates per film type were obtained.

##### Antibacterial Activity Analysis

The agar disk diffusion method was employed to determine antibacterial activities. *Listeria monocytogenes* (CMCC 54004), *Staphylococcus aureus* (CMCC 26112), *Salmonella* Enteritidis (CMCC 50071), and *Escherichia coli* (CMCC 44113) were used, with 100 μL containing 10^7^ colony-forming units per mL (CFU/mL) smeared onto the surface of each Mueller-Hinton agar plate (Aladdin Reagent Co. Shanghai, China). A 10 mm diameter sample of the film was obtained with a hole punch (Deli Co. Ningbo, Zhejiang, China) and placed on the inoculated agar plates. Finally, the plates were incubated at 37 °C for 24 h, and the diameters (mm) of the zones of inhibition were measured using a caliper (Deli Co. Ningbo, Zhejiang, China). The tests were performed in triplicate. 

### 4.5. Meat Quality Assessment

#### 4.5.1. Preparation of Meat Samples and Treatments

Fresh beef was obtained from a local market (Harbin, China) and quickly transported to the laboratory where it was sliced using sterile cutting boards and knives on an aseptic operating table. Samples (approximately 50 g without bones) were randomly divided into six groups. The meat samples were dipped in coating solutions for 30 s, drained for 30 min, placed in polystyrene trays sealed with PE films, and then stored at 4 ± 1 °C for treatment.

#### 4.5.2. Measurement of Lipid Oxidation

The lipid oxidation of meat was calculated using thiobarbituric acid reactive substances (TBARS) according to a reported method with slight modifications [41]. A 10 g sample was dispersed in 50 mL 15% trichloroacetic acid (TCA) and then homogenized for 2 min. The homogenate was centrifuged at 5900× *g* for 5 min at 4 °C (Z236HK Hermle, Wehingen, Germany). The supernatant was vacuum filtered through a fiber membrane (0.45 µm pore size, Aladdin, Shanghai, China). The filtrate (4 mL) was then mixed with 0.06 M thiobarbituric acid (TBA) using a vortexer (Jia Jun bio-technology Co., Nanjing, China) for 15 s, incubated in a water bath at 90 °C for 90 min, and cooled in an ice bath. Absorbance was measured at 520 nm on a spectrophotometer (UV-2500, Shimadzu Co, Kyoto, Japan). The concentration of malonaldehyde was calculated based on a standard curve of 1, 1, 3, 3-tetramethoxypropane (TMP) and expressed as mg MDA/kg of sample.

#### 4.5.3. Measurement of Antioxidant Activity with DPPH Radical-Scavenging Activity

The DPPH radical-scavenging activity was determined according to a previously described procedure [27]. A meat sample was mixed with methanol (1:10 w/v), homogenized for 1 h, and then centrifuged at 1500× *g* for 10 min at 4 °C. One milliliter of sample solution was mixed with 5.0 mL DPPH (0.004% in methanol). The absorbance of the mixture was measured at 517 nm with a spectrophotometer (UV-2500, Shimadzu Co, Kyoto, Japan) after 30 min in the dark. Vc was used as the positive control. The DPPH (%) radical-scavenging activity was calculated according to the following equation:$$\mathrm{Scavenging}\;\mathrm{rate}(\%)=\frac{A_{0}-A_{i}}{A_{0}}\times 100$$where *A*_i_ is the absorbance of the sample, and *A*_0_ is the absorbance of the control sample.

#### 4.5.4. Microbiological Analyses

Microbiological analysis was performed as described elsewhere [42] with some modifications. The meat samples (20 g) were mixed with 200 mL of peptone saline solution (8.5 g/L NaCl and 1 g/L peptone) and homogenized using a stomacher 400 (Seward Medical, London, UK) for 2 min. One hundred microliters of each dilution was poured onto an agar plate and then incubated at 4 °C for 15 days. During the storage period, total viable counts (TVC) were determined using plate count agar every 3 days.

#### 4.5.5. Weight Loss

Water loss is the main reason for the loss of weight in fresh meat. Weight loss was considered the difference between the initial meat weight (*W*_initial_) and the meat weight at each time point (*W*_final_), which was expressed as a percentage as follows:$$\mathrm{Weight}\;\mathrm{loss}\%=\frac{W_{\mathrm{initial}}-W_{\mathrm{final}}}{W_{\mathrm{initial}}}\times 100$$

### 4.6. Statistical Analysis

All treatments were performed in triplicate. In the table and figure, the values are the means of the triplicate measurements, and error bars indicate the standard deviations. A statistical analysis was performed using SPSS (20.0) software. Significant differences (*p* < 0.05) between means were identified using Duncan’s multiple range test.

## 5. Conclusions

WPNFs can be used as a source of protein-based biodegradable films and coatings, relying on their well-ordered *β*-sheet structures, higher hydrophobicity, homogeneous structure, and antioxidant activity. TNTs with excellent photo catalytic properties can serve as antimicrobial agents when added to EFCs. The good interactions between TNTs and WPNFs gave the films a smooth and continuous surface and presented a homogeneous, compact, and cohesive interior, with good mechanical properties. The higher absorbance of the WPNF/TNT films in the UV region suggests that they can prevent lipid oxidation and food deterioration. Combined with TBARS values and DPPH radical-scavenging activities, these data suggest that WPNFs as ECs can help limit lipid peroxidation and promote antioxidant activity in beef. The antimicrobial activities and TVCs indicated that the addition of TNTs can reduce microbial growth and extend the shelf life of chilled beef. WPNF/TNT ECs also led to a significant reduction in weight loss. Given that the WPNF/TNT film components are low cost and showed high antioxidant and antimicrobial activity, such optimized films show great potential for applications to various food products, including raw and chilled meat.

## Figures and Tables

**Figure 1 ijms-20-01184-f001:**
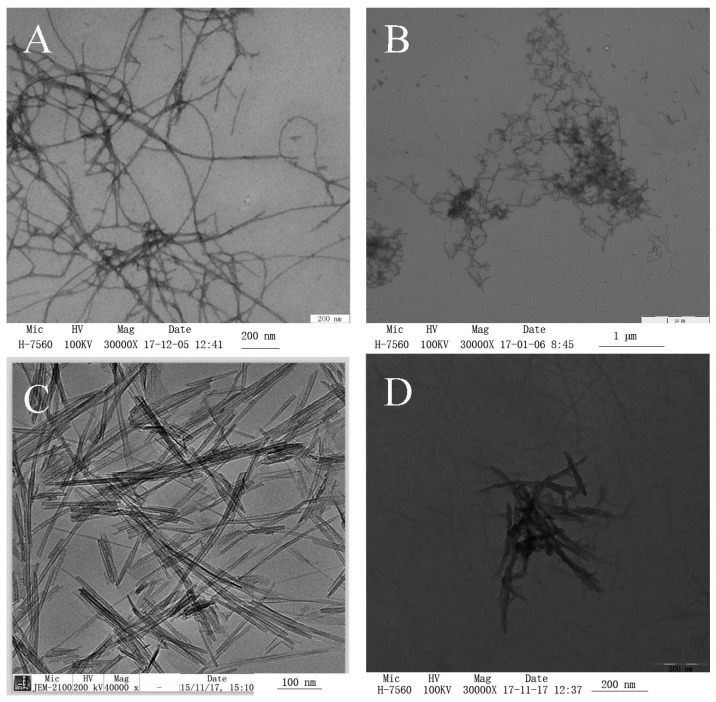
TEM micrographs (negatively stained) of the coating solution. (**A**) WPNFs at pH 2.0, (**B**) WPNFs at pH 7.0, (**C**) TNTs (TEM micrographs provided by Beijing Dk Nano S&T, Ltd. [Beijing, China]), (**D**) WPNF-TNTs at pH 7.0.

**Figure 2 ijms-20-01184-f002:**
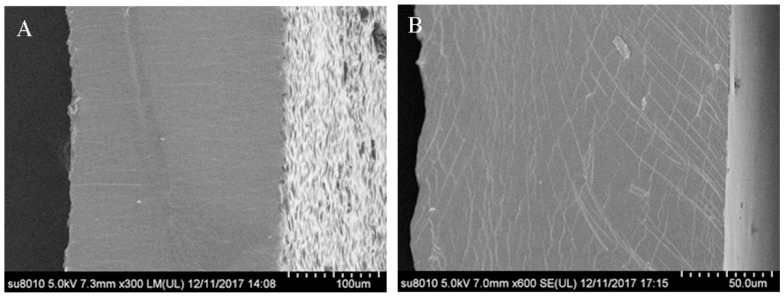
SEM micrographs of the surfaces and cross-sections of the composite films. (**A**) WPI/TNT films and (**B**) WPNF/TNT films.

**Figure 3 ijms-20-01184-f003:**
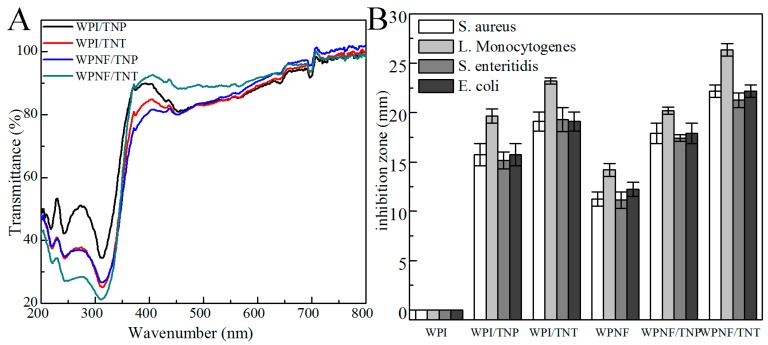
Properties of the nano-composite films. (**A**) Variations in the transmittance (T) of the nano-composite films in the range of 200–800 nm. (**B**) Antimicrobial activity of nanocomposite films.

**Figure 4 ijms-20-01184-f004:**
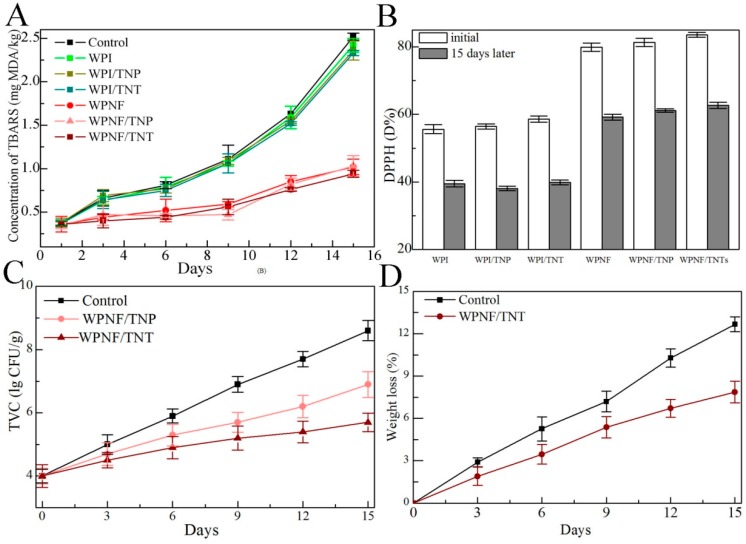
Functional properties of the ECs on the preservation of chilled meat. (**A**) Lipid oxidation effects of edible coatings for meat samples measured by TBARS concentrations during storage. (**B**) Antioxidant effects of edible films against DPPH radicals (D%) by storage day. (**C**) Total viable counts of meat during storage at 4 °C. (**D**) Weight loss (%) of coated and uncoated meat.

**Table 1 ijms-20-01184-t001:** Thickness, tensile strength (TS), and elongation at break (EB) of films.

Film	Thickness (mm)	TS (MPa)	EB (%)
WPI	0.147 ± 0.023^a^	9.53 ± 0.33^a^	4.95 ± 0.07^b^
WPI/TNPs	0.139 ± 0.037^a^	11.42 ± 0.33^c^	4.26 ± 0.07^a^
WPI/TNTs	0.141 ± 0.042^a^	13.45 ± 0.41^d^	4.96 ± 0.66^b^
WPNFs	0.142 ± 0.034^a^	10.49 ± 0.35^b^	5.88 ± 0.66^d^
WPNFs/ TNPs	0.144 ± 0.035^a^	14.73 ± 0.31^e^	5.36 ± 0.70^c^
WPNFs/TNTs	0.147 ± 0.051^a^	15.49 ± 0.35^f^	5.83 ± 0.09^d^

Mean values with standard deviations. Different letters (a–f) within the same column indicate significant differences between the films (*p* < 0.05).

**Table 2 ijms-20-01184-t002:** Transparency, moisture content, and water solubility of films.

Film	MC (%)	Solubility in Water (%)	Transparency (%)	WVP (10 ^−11^gm/m^2^sPa)
WPI	33.1 ± 1.1^d^	42.2 ± 0.9^f^	30.4 ± 1.1^c^	8.1 ± 0.21^f^
WPI/TNP	34.3 ± 1.1^e^	35.2 ± 1.3^e^	25.4 ± 2.1^a^	5.7 ± 0.21^d^
WPI/TNT	35.6 ± 1.7^f^	34.6 ± 1.4^d^	27.5 ± 2.2^b^	4.9 ± 0.32^c^
WPNF	24.1 ± 1.1^a^	29.2 ± 1.3^c^	53.1 ± 0.9^f^	6.2 ± 0.32^e^
WPNF/TNP	27.6 ± 2.1^c^	27.6 ± 2.1^b^	40.1 ± 3.1^d^	4.4 ± 0.22^b^
WPNF/TNT	24.3 ± 0.7^b^	26.3 ± 1.5^a^	48.6 ± 1.2^e^	3.7 ± 0.13^a^

Mean values with standard deviations. Different letters (a–f) within the same column indicate significant differences between the films (*p* < 0.05).

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
