# Peer review of "Effect of Antioxidant and Antimicrobial Coating based on Whey Protein Nanofibrils with TiO2 Nanotubes on the Quality and Shelf Life of Chilled Meat"

_ijms, 2019, doi:10.3390/ijms20051184_

Round 1

Reviewer 1 Report

Research is very extensive and gives significant impact on gaining new knowledge about possibilities and application of whey protein nanofibrils and titanium dioxide containing packaging materials for various food products, including raw and chilled meat.

 Introduction gives short look into reported protein based edible films and their advantages as well as about titanium dioxide as antimicrobial agent. In comparison with other quite excellent parts of manuscript, introduction part could be improved with additional information. Manuscript contains very detailed description of used materials and methods. Results are explained and discussed in details and presented in several images as well as in tables.

Manuscript is very well organized, conclusions present essence of results and discussion parts.

 I suggest to publish manuscript after minor revision, however I have comments:

1. Fig1. It would be better to have clear scale ruler on every image with clearly readable units. As language of manuscript is English, I don’t see necessity to display subscriptions in other language under the TEM pictures- there are one line under each TEM image.

2. My suggestion is to reorganize some subtitles in Results part of manuscript, they repeat the same information as title. For instance,

2.2 Characterization and physical properties of films

2.2.1 Scanning electron microscopy (SEM) of the microstructure of films

2.2.3 Physical properties and characterization of nanocomposite films

Author Response

Thank you very much for taking time to review our manuscript. We thank you for your thoughtful suggestions and insights, which have enriched the manuscript and produced better account of the research.

Point 1: Fig1. It would be better to have clear scale ruler on every image with clearly readable units. As language of manuscript is English, I don’t see necessity to display subscriptions in other language under the TEM pictures. there are one line under each TEM image.

Response: We have revised the picture Fig.1 according to the reviewer’s comments.

Point 2: My suggestion is to reorganize some subtitles in Results part of manuscript, they repeat the same information as title. For instance,

2.2 Characterization and physical properties of films

2.2.1 Scanning electron microscopy (SEM) of the microstructure of films

2.2.3 Physical properties and characterization of nanocomposite films

Response: We have revised the subtitles in results part of our manuscript according to the reviewer’s comments on page 3 lines 74 and page 4 lines 102.

Reviewer 2 Report

Manuscript Number: ijms-453589

This manuscript reported an effect of the antioxidant and antimicrobial coating based on whey protein nanofibrils with TiO2 nanotubes on the quality and shelf life of chilled meat. The results provided are good to have a novel and effective strategy for use in food packaging systems to
prevent spoilage from light-induced oxidation. However, the authors fail to explain the importance and actual role of
TiO2 in antibacterial activities. Also, it is more curious to explain state-of-art. An author should provide comparative results with pure TNTs. In addition, need to improve the discussion part very precisely and clearly.

Author Response

Thank you very much for taking time to review our manuscript. We thank you for your thoughtful suggestions and insights, which have enriched the manuscript and produced better account of the research.

Point 1: This manuscript reported an effect of the antioxidant and antimicrobial coating based on whey protein nanofibrils with TiO2 nanotubes on the quality and shelf life of chilled meat. The results provided are good to have a novel and effective strategy for use in food packaging systems toprevent spoilage from light-induced oxidation.

Response 1: Thank you very much for taking time to review our manuscript. The affirmation on our work has greatly encouraged us.

Point 2: However, the authors fail to explain the importance and actual role of TiO2 in antibacterial activities. Also, it is more curious to explain state-of-art.

Response 2: We have added some explanations about the importance and actual role of TiO2 in antibacterial activities on page 8 lines 233-238.

Point 3: An author should provide comparative results with pure TNTs.

Response 3: This is still a very considerate suggestion, which we will consider it in our future experiments. TNTs are difficult to disperse uniformly in water, and WPI and WPNFs can help TNTs be dispersed uniformly. The antimicrobial activity of TNTs is well known, and compare with WPI and WPNFs we also can know the contribution of TNTs.

Point 4: In addition, need to improve the discussion part very precisely and clearly.

Response 4: We have improved the discussion part very precisely and clearly on page 8 lines 218-page 9  lines 274.